# Influence of an Overshoot Layer on the Morphological, Structural, Strain, and Transport Properties of InAs Quantum Wells

**DOI:** 10.3390/nano14070592

**Published:** 2024-03-27

**Authors:** Omer Arif, Laura Canal, Elena Ferrari, Claudio Ferrari, Laura Lazzarini, Lucia Nasi, Alessandro Paghi, Stefan Heun, Lucia Sorba

**Affiliations:** 1NEST, Istituto Nanoscienze-CNR and Scuola Normale Superiore, Piazza S. Silvestro 12, I-56127 Pisa, Italy; omer.arif@sns.it (O.A.); l.canal@studenti.unipi.it (L.C.); alessandro.paghi@nano.cnr.it (A.P.); stefan.heun@nano.cnr.it (S.H.); 2Istituto dei Materiali per l’Elettronica ed il Magnetismo, Consiglio Nazionale delle Ricerche (IMEM–CNR), Parco Area delle Scienze 37/A, I-43124 Parma, Italy; elena.ferrari@imem.cnr.it (E.F.); claudio.ferrari@imem.cnr.it (C.F.); laura.lazzarini@imem.cnr.it (L.L.); lucia.nasi@imem.cnr.it (L.N.)

**Keywords:** molecular beam epitaxy, quantum wells, metamorphic buffer layers, InAs, strain, semiconductors, III–V materials, structural properties, high-resolution X-ray diffraction, electrical properties

## Abstract

InAs quantum wells (QWs) are promising material systems due to their small effective mass, narrow bandgap, strong spin–orbit coupling, large g-factor, and transparent interface to superconductors. Therefore, they are promising candidates for the implementation of topological superconducting states. Despite this potential, the growth of InAs QWs with high crystal quality and well-controlled morphology remains challenging. Adding an overshoot layer at the end of the metamorphic buffer layer, i.e., a layer with a slightly larger lattice constant than the active region of the device, helps to overcome the residual strain and provides optimally relaxed lattice parameters for the QW. In this work, we systematically investigated the influence of overshoot layer thickness on the morphological, structural, strain, and transport properties of undoped InAs QWs on GaAs(100) substrates. Transmission electron microscopy reveals that the metamorphic buffer layer, which includes the overshoot layer, provides a misfit dislocation-free InAs QW active region. Moreover, the residual strain in the active region is compressive in the sample with a 200 nm-thick overshoot layer but tensile in samples with an overshoot layer thicker than 200 nm, and it saturates to a constant value for overshoot layer thicknesses above 350 nm. We found that electron mobility does not depend on the crystallographic directions. A maximum electron mobility of 6.07 × 10^5^ cm^2^/Vs at 2.6 K with a carrier concentration of 2.31 × 10^11^ cm^−2^ in the sample with a 400 nm-thick overshoot layer has been obtained.

## 1. Introduction

InAs and high In-content InGaAs/InAlAs quantum wells (QWs) are attractive material systems due to their small effective mass, narrow bandgap, strong spin–orbit coupling, large g-factor, and transmissive interface to superconductors [1,2,3,4,5]. Recent studies have shown that InAs is an excellent candidate for high-speed electronic devices, optoelectronics, spintronics, quantum computing, and topological superconducting circuits [4,5,6,7,8,9,10,11,12]. Furthermore, InAs-based two-dimensional electron gases (2DEGs) with an in situ deposited Al epitaxial layer led to the observation of conductance quantization in quantum point contacts [4,11], to the realization of Josephson junctions with tunable supercurrent [11], and the observation of Andreev bound states that may host Majorana zero modes [4,12].

One of the fundamental issues of scalability with the growth of high-quality and high-mobility InAs and high In-content InGaAs/InAlAs QWs is the large lattice mismatch with commonly available substrates. The only lattice-matched substrates for growing InAs and high In-content InGaAs/InAlAs QWs are InAs and GaSb. Since InAs substrates are conductive, they are not suitable for electronic applications. On the other hand, GaSb substrates are very expensive, and tellurium rather than silicon must be used to dope the AlGaSb barriers [13,14]. To overcome these issues, researchers have implemented step-graded metamorphic buffer layers (MBLs) of InGaAs or InAlAs on GaAs and InP substrates [3,4,5,15,16,17,18,19,20]. Step-graded MBLs allow the achievement of virtually strain-free and dislocation-free QWs by producing misfit dislocations at compositional steps in the metamorphic buffer layer far from the active region of the structure. Recently, Benali et al. [16] and Hatke et al. [4] reported the highest-ever mobility of 7 × 10^5^ cm^2^/Vs and 1 × 10^6^ cm^2^/Vs at 4.2 K for InAs/InGaAs QWs on GaAs(001) and InP(100) substrates, respectively. For quantum computing and superconducting applications, GaAs substrates are more attractive due to the high resistivity shown and, therefore, this work has been devoted to the study of MBLs on GaAs substrates.

High electron mobility is an important requisite to investigate the fundamental properties of materials. Therefore, it is crucial to determine the factors limiting electron mobility. A lot of work has already been performed, and in this process, the properties of these materials [3,4,5,15,16,17,18,19,20] have been improved. Ionized impurity and alloy disorder scattering are the main scattering mechanisms affecting mobility at low temperatures [3,4,15,21]. Recently, Hatke [4] et al. and Shabani et al. [22] experimentally confirmed that unintentional impurities limit the mobility of undoped InAs/InGaAs QWs. Another interesting result is that InAs/InGaAs and InGaAs QWs grown on standard GaAs(100) wafers are reported to show an anisotropy in mobility [3,5,20,21,23]. The mobility was found to be higher along the $\left[1\bar{1}0\right]$ direction compared to the [110] direction [3,20,21,23]. This effect was attributed to several reasons, such as interface roughness, indium concentration modulation, asymmetric strain relaxation, anisotropic ordering effects, and anisotropic growth of dislocations [3,5,20,21,23,24].

In this work, we investigate undoped InAs/In_0.81_Ga_0.19_As QWs grown using solid source Molecular Beam Epitaxy (MBE) on semi-insulating GaAs(100) substrates. We systematically investigated their morphological, structural, strain, and transport properties. First, we studied the evolution of surface morphology by performing atomic force microscopy (AFM). Then, a study of the structural quality was performed in cross-section using transmission electron microscopy (TEM). The residual strain was determined using X-ray diffraction (XRD) rocking curves and reciprocal space maps (RSMs). To derive the sheet carrier density and mobility of samples at 4.2 K, we first performed Hall measurements in the van der Pauw configuration. Then, the electron mobility in $\left[1\bar{1}0\right]$ and [110] directions was determined via Hall bars (HBs) at 2.6 K. This study allowed us to demonstrate that optimizing the residual strain of the buffer layer provides an efficient way to achieve excellent electrical properties of InAs 2DEGs. Moreover, within error bars, the electron mobility is independent of crystallographic directions due to the strong confinement of the wave function of the carriers inside the binary InAs QW.

## 2. Experimental Details

The QWs studied in this paper were grown on semi-insulating GaAs(100) substrates via solid source Molecular Beam Epitaxy (MBE) in a Riber compact 21 DZ system (Riber, Paris, France). A schematic of our heterostructure adapted from Refs. [15,16] is shown in Figure 1. The sequence of the layer structure in all samples, starting from the substrate, is as follows: a 200 nm GaAs layer, a 100 nm GaAs/AlGaAs superlattice (SL), a 200 nm GaAs layer, a 1250 nm step-graded In_x_Al_1−x_As metamorphic buffer layer with x increasing from 0.15 to 0.81, an overshoot layer with In concentration x = 0.84, and an active region with In concentration x = 0.81. The buffer layer consists of two regions with different misfit gradients df/dt. The first region is composed of twelve 50 nm steps with x ramping from 0.15 to 0.58, corresponding to a misfit gradient of 5.1% μm^−1^. The second region is composed of twelve 50 nm steps with x ramping from 0.58 to 0.81, corresponding to a misfit gradient around 3.1% μm^−1^. The flux of Al was kept constant during the buffer layer growth, while the In cell temperature was increased at each step without growth interruptions. The active part of the sample from the buffer layer to the free surface is composed of a 50 nm In_0.81_Al_0.19_As barrier, a 9 nm In_0.81_Ga_0.19_As layer, a 7 nm InAs QW, a 9 nm In_0.81_Ga_0.19_As layer, a 117 nm In_0.81_Al_0.19_As barrier, and a 3 nm In_0.81_Ga_0.19_As capping layer. The buffer layer and the active part were grown at optimized substrate temperatures of (320 ± 5) °C and (480 ± 5) °C, respectively. The arsenic flux was adjusted during the growth to keep a group V/III beam flux ratio of 8 throughout the growth. A growth rate of about 1.3 μm/h was used for the active region. Although nominally undoped, electronic charge in the quantum well is due to a deep-level donor state in the InAlAs barrier band gap, whose energy lies within the InGaAs/InAlAs conduction band discontinuity [25]. A series of samples with different overshoot layer thicknesses was grown to study the effect of strain relaxation on the transport properties of the InAs QWs. The samples presented in this study have different overshoot layer thicknesses (t_OS_) of 200 nm, 250 nm, 300 nm, 350 nm, and 400 nm and are labeled as A, B, C, D, and E, respectively.

The surface morphology of the grown samples was studied using a Bruker Dimension Icon atomic force microscope (AFM) (Bruker, Köln, Germany). The scans were performed in the air using the ScanAsyst operative mode (NanoScope, 9.1), which is a non-resonance tapping mode. The tip is made of silicon nitride and has a radius of < 10 nm and a height of 2.5–8 µm. To investigate different regions of the sample, we performed scans of 10 × 10 μm^2^, 20 × 20 μm^2^, and 50 × 50 μm^2^ areas.

Samples for TEM observations were prepared in both plan-view and cross-section using the standard mechano-chemical thinning procedure. The useful thickness for electron transparency was obtained with final thinning in the GATAN 691 Precision Ion Polishing System (PIPS) (Pleasanton, CA, USA). The samples were observed in both conventional TEM and High-Angle Annular Dark-Field scanning TEM (HAADF-STEM) imaging modes in an analytical JEOL 2200FS UHR (Tokyo, Japan) field emission microscope, using standard rules for dislocation contrast optimization.

X-ray diffraction (XRD) measurements were performed to investigate the In concentration and residual strain in the QWs using a QC3 Bruker (Bruker, Migdal Ha’Emek, Israel) high-resolution diffractometer. The X-ray tube consists of a tungsten filament and a copper anode, operating at a voltage of 40 kV and a current of 40 mA. The X-ray optics consist of a parabolic graded multilayer mirror and a channel-cut collimator with a 2-bounce Ge(004) crystal. The parabolic mirror parallelizes the beam with a divergence of ~0.1°, while the 2-bounce Ge(004) monochromator selects the Cu K_α1_ wavelength (1.54059 Å). Symmetric (004) and asymmetric (115) ω-2θ scans were performed to calculate the out-of-plane and in-plane lattice constants of lattice-mismatched epilayers grown on top of the GaAs(100) substrate. Reciprocal space maps (RSMs) of sample E were acquired with a Philips (Amsterdam, Netherlands) X-Pert high-resolution XRD equipped with a Bartels (Hamburg, Germany) four-crystal Ge(220) monochromator, using Cu K_α1_ radiation with a 12 arcsec angular divergence. We have measured reciprocal lattice maps of (004) and (224) diffraction along [110] and $\left[1\bar{1}0\right]$ scattering planes by setting the azimuthal angle φ to angles 0°/180° and 90°/270°.

Transport measurements of all samples were performed at 4.2 K on ~4 × 4 mm^2^ samples in van der Pauw geometry. A Hall-bar (HB) geometry was used to determine if any anisotropic mobility was present in sample E (the sample with the highest electron mobility) at 2.3 K. HBs of dimensions (L × W) = 550 × 100 µm^2^ were fabricated along the $\left[1\bar{1}0\right]$ and [110] directions via optical lithography and wet etching. Here, L is the length, and W is the width of the HB. Electrical contact to the 2DEG is guaranteed by alloyed Ni/Ge/Au Ohmic contacts. The distance between the longitudinal contacts was 250 µm.

## 3. Results and Discussion

The surface morphology of all samples with different overshoot layer thicknesses was investigated using AFM. The AFM micrographs were acquired on the free surface of the samples (the capping layer of the samples), which reflects the morphology of the buried In_x_Al_1−x_As/In_x_Ga_1−x_As interfaces. As an example, we show AFM micrographs of samples A (200 nm-thick overshoot layer) and E (400 nm-thick overshoot layer) and corresponding line scans shown in Figure 2. All samples show a cross-hatched pattern oriented along the two orthogonal directions $\left[1\bar{1}0\right]$ and [110]. The cross-hatched patterns are periodic undulations of surface morphology typical of lattice-mismatched heteroepitaxy. Such striations derive from an inhomogeneous strain relaxation process caused by interfacial misfit dislocations buried in the buffer layer [5,15,18,26]. For all samples, we find a similar two-dimensional root mean square (RMS) roughness of the sample surface, with a value of (3.2 ± 1) nm. The RMS value given above is the average from three images of size 20 × 20 μm^2^ acquired from different regions of each sample. Furthermore, we obtained the one-dimensional RMS roughness of the samples along the $\left[1\bar{1}0\right]$ and [110] directions by analyzing line scans of 10 µm. The RMS roughness of sample A along the $\left[1\bar{1}0\right]$ and [110] directions is (1.90 ± 0.3) nm and (2.50 ± 0.6) nm, respectively, while for sample E, it is (1.86 ± 0.3) nm and (1.92 ± 0.6) nm, respectively. Thus, no significant difference between the two orthogonal crystallographic directions was observed. On the other hand, the period of the surface oscillations does depend on the crystallographic direction. It is shorter in the [110] direction and longer in the $\left[1\bar{1}0\right]$ direction. The average period of undulations for samples A and E in the [110] direction is (540 ± 140) nm and (505 ± 70) nm, respectively. In the $\left[1\bar{1}0\right]$ direction, the average period of undulations for samples A and E is (1340 ± 225) nm and (1470 ± 250) nm, respectively. These AFM findings agree well with prior studies [15,16,18].

The distribution of dislocations across the metamorphic buffer layer was studied using TEM on sample C (300 nm overshoot layer thickness) and sample E (400 nm overshoot layer thickness), both in plan view and cross-section. The study of the planar sections was carried out in areas of tens of square microns throughout the electron-transparent zone (the maximum thickness for reliable observation is equal to approximately 500 nm from the sample surface). No misfit dislocations were observed in both samples. However, occasional threading dislocations emerging at the surface were observed, as shown in Figure 3a. Their density was estimated to be approximately 10^7^ cm^−2^, without notable differences between the two samples. The poor statistics intrinsic to the technique make this estimate less precise. Figure 3a,b show cross-sectional dark field Scanning TEM images of samples C and E, respectively, taken along a <110> axis. TEM images were captured in areas up to 10 µm in length. The different regions of the layer structure are marked in the images. Two regimes are clearly observed in both samples: a region close to the substrate with a high density of misfit dislocations and a region near the surface where no misfit dislocations are observed, consistent with the plan-view investigations. All misfit dislocations are confined within the graded metamorphic buffer layer. The distance measured from the surface to the last misfit dislocation layer for samples C and E is (505 ± 10) nm and (625 ± 5) nm, respectively. The difference between the two samples is due to their different overshoot layer thickness, which, thus, does not contain any misfit dislocations. These results confirm that, by carefully choosing the design of the metamorphic buffer layer, the active region can be grown free from misfit dislocations.

Figure 4a shows a high-resolution STEM-HAADF image of the In_0.81_Ga_0.19_As/InAs/In_0.81_Ga_0.19_As layers obtained from sample E. The InAs QW is indicated in the image. The InAs layer appears brighter and more clearly visible than the In_0.81_Ga_0.19_As layers due to its higher average atomic number with respect to the barriers. This allowed us to determine the thickness of the InAs QW as (7.5 ± 0.5) nm, which is in good agreement with the nominal thickness. The energy dispersive X-ray spectroscopy (EDS) line scan analysis using the K_α1_ emission of Ga and the L_α1_ emission of In along the red line marked in panel (a) confirmed that Ga is present only in the two layers bordering the InAs QW (Figure 4b). The broadening of the signal is due to the size of the characteristic X-ray generation volume, which is larger than the InAs QW.

To measure the residual strain in the samples, we performed XRD measurements in triple-axis configuration along 0° and 180° azimuth in the vicinity of the (004) reflection. Figure 5a compares the ω-2θ scans of the series of samples with increasing overshoot layer thickness, measured under 0° azimuth. All spectra display three main peaks: the substrate peak, the active region (AR) peak, and the overshoot (OS) layer peak. The peak with the highest intensity is the substrate peak. The broad features in the range from -7000 arcsec to -1500 arcsec originate from the metamorphic buffer layer. Around -5000 arcsec, there is a small peak corresponding to an indium content of 0.58, which is associated with the accumulation of an In_x_Al_1−x_As layer with x = 0.58, at which the slope of the indium profile in the structure changes. The similar position of this peak for all samples indicates that the buffer layer profiles are the same. Therefore, the changes in the strain relaxation of the active region can be uniquely associated with the differences in the thickness of the overshoot layers.

We observed that both the AR and the OS peaks shift to the right with increasing t_OS_ due to different strain relaxation in these layers. We calculated the residual perpendicular strain in the active region and in the overshoot layer, assuming the nominal composition and using Vegard’s law. The residual strain of the samples is plotted in Figure 5b as a function of overshoot thickness t_OS_. The overshoot layers are compressively strained for all t_OS_, but with increasing t_OS_, the strain decreases. Instead, the strain in the active region switches from compressive to tensile and tends to saturate above 350 nm, where the overshoot layer is nearly relaxed. In summary, by varying the thickness of the overshoot layer, it is possible to tune the strain in the active region to achieve a virtual substrate that is lattice-matched to the active region.

Sample E is further investigated by measuring RSMs near symmetric (004) and asymmetric (224) nodes, with the diffracting plane inclined 35.26° with respect to the (001) nominal surface. The residual strain was measured in both $\left[1\bar{1}0\right]$ and [110] directions by setting the azimuthal angle φ to angles 0°/180° and 90°/270°, respectively, corresponding to (2−24), (224), (−224), and (−2−24) diffracting planes. Figure 6 shows symmetric (004) RSMs of sample E with scattering planes parallel to $\left[1\bar{1}0\right]$ and [110] directions. All maps show three different intense spots corresponding to the substrate, the AR, and the OS. The AR and OS peaks appear shifted with respect to the position of the substrate peak due to (a) the presence of small unintentional miscuts (α(1−10) = 0.32°; α (110) = 0.13°) of the crystal surface with respect to the (001) nominal orientation, and (b) due to lattice tilts of OS and AR with respect to the substrate lattice (α (1−10) = 0.10°; α (110) = 0.27°), due to unbalanced Burgers vector components of misfit dislocations perpendicular to the (001) surface [27]. Since such dislocations are buried in the buffer layer, the tilt is the same for both OS and AR. We also note larger mosaic spreads of OS and AR peaks in maps parallel to the [110] direction. This is an indication of a higher dislocation density, and hence a higher strain release, along the [110] direction. Consistent with this [28,29], the lattice tilt is also larger along this direction.

We assess the strain state of the structure from the RSMs near the 224 asymmetric nodes. Figure 7 shows asymmetric (224) RSMs of sample E measured at azimuthal angles φ = 0° and φ = 90°, around (2–24) and (224) reciprocal lattice vectors, respectively. Figure 7 (left picture) shows the AR peak aligned along the (224) reciprocal lattice vector of the GaAs substrate, indicating a strain-free AR layer in the [110] direction. We also observe that (224) AR and OS peaks are vertically aligned, indicating a parallel lattice match between the AR and OS layers. We thus conclude that the OS layer is in a compressive state in this direction.

We obtained the parallel and perpendicular lattice parameters *a^par^* and *a^perp^* from the *k_x_* (=*k^par^*) and *k_y_* (=*k^perp^*) positions of the OS and AR peaks in the RSMs. The parallel and perpendicular strain ε*^par^* and ε*^perp^* of AR and OS layers can then be calculated as *ε ^par/perp^ = (a ^par/perp^−a^relaxed^)/a^relaxed^*. The substrate miscut and the epilayer tilt were corrected for by averaging the peak positions between RSMs measured with azimuthal angles differing by 180°. We calculated the relaxed lattice parameters using the InAs Poisson ratio ν = 0.352, according to *a^relaxed^* = *a^perp^* × (1 + *ν*)/(1 − *ν*) + *a^par^* × 2*ν*/(1 − *ν*), where *a^par^* was averaged between the [110] and $\left[1\bar{1}0\right]$ directions. We also estimated the In fraction in both OS and AR following Vegard’s law. The results are consistent with the nominal values. The data are summarized in Table 1.

Hence, anisotropy is observed in the strain relaxation of the AR region, with negligible strain along the $\left[1\bar{1}0\right]$ direction and tensile strain along the [110] direction. The *ε^perp^* value of the AR (–0.0016) agrees with the data reported in Figure 5b. This anisotropy correlates with the higher dislocation density along the [110] direction, corresponding to more misfit dislocation lines parallel to the $\left[1\bar{1}0\right]$ direction [30,31,32]. Two types of perfect dislocations with Burgers vector b = a/2(110) exist in III–V compound semiconductors, known as α and β types. The nucleation and gliding of α and β dislocations control the relaxation along the [110] and $\left[1\bar{1}0\right]$ directions. Due to the different core structures of these dislocations, there is a significant difference in activation energy for dislocation nucleation and glide. This behavior of dislocation motion is responsible for the observed strain relaxation anisotropy.

We determine the two-dimensional (2D) electron density (N_s_) and mobility (µ) of all samples, as shown in Figure 8. Samples are measured in a van der Pauw configuration at 4.2 K. N_s_ is nearly independent of overshoot layer thickness and lies between 2.3 × 10^11^ cm^−2^ and 2.7 × 10^11^ cm^−2^. On the other hand, the mobility of samples A, B, C, and D lies in the range of 3.4 × 10^5^ cm^2^/Vs to 4.4 × 10^5^ cm^2^/Vs. Sample E has a peak mobility of 5.55 × 10^5^ cm^2^/Vs with a carrier concentration of 2.4 × 10^11^ cm^−2^.

The main scattering mechanisms limiting the mobility in undoped InAs/InGaAs QWs are background impurities, interface roughness, and alloy disorder scattering [3,4,16,21]. The most important scattering mechanism in high-mobility QWs is background impurities scattering [4,16]. However, the concentration of background impurities is similar for all samples discussed here because it depends on the MBE system. Furthermore, alloy disorder scattering is comparable for all samples because it is related to the disorder in the ternary alloys of the barriers, which have the same thickness and composition in all samples. The AFM study revealed that all samples show similar RMS roughness. Thus, surface roughness is not the cause of the observed behavior of mobility.

A possible explanation for the lower mobility of samples A to D compared to sample E could be the smaller distance between the QW region and the last misfit dislocations in the buffer layer. The values measured using TEM are (375 ± 10) nm for sample C and (495 ± 5) nm for sample E. In fact, it is well known that dislocations can act as trapping centers and thus create a random electric field inside the material [15,33]. The smaller distance of this random electric field to the well in samples A to D, with respect to sample E, could be a possible cause of the mobility reduction.

Another difference between the samples of the series is the residual strain in the active region layers. Theoretical work has demonstrated that strain affects the mobility of lattice-mismatched QWs due to random fluctuations in the strain field, which produce a fluctuating piezoelectric field [34]. Sample A has a compressively strained active region with the out-of-plane lattice constant of the InAlAs barriers larger than the in-plane lattice constant. Therefore, the pseudomorphically grown InAs QW is compressively strained. Instead, the active regions of the other samples are subject to a tensile strain due to the presence of a thicker and more relaxed overshoot layer. There seems to be, however, no clear inverse correlation between the strain in the active region and the mobility: while samples A and B have the lowest absolute values of strain in the active region, it is sample E that has the highest mobility, despite some tensile strain in the active region. This indicates that a certain amount of tensile strain in the QW promotes mobility. Indeed, it is well known that the bandgap of InAs reduces under the influence of tensile strain [35,36]. Strain also affects the effective mass of the material [35,36]; due to the reduction of bandgap, the effective mass also decreases. Since carrier mobility is inversely proportional to the carrier’s effective masses, a reduction in the effective mass will lead to an increase in mobility.

A possible anisotropy in mobility is investigated by fabricating Hall-bar devices on the highest mobility sample E. Specifically, HBs were aligned along the $\left[1\bar{1}0\right]$ and [110] directions. N_s_ and µ were measured at 2.6 K via the Hall effect. We measured four HBs in both crystallographic directions. The measured average value of N_s_ along the [110] direction is (2.33 ± 0.08) × 10^11^ cm^−2^, while along the $\left[1\bar{1}0\right]$ direction, it is (2.28 ± 0.03) × 10^11^ cm^−2^, in very good agreement with the value of sample E obtained with the van der Pauw method (see Figure 8). On the other hand, the measured average value of µ along the [110] direction is (6.09 ± 0.57) × 10^5^ cm^2^/Vs, while along the $\left[1\bar{1}0\right]$ direction, it is (6.04 ± 0.23) × 10^5^ cm^2^/Vs. Within error bars, the 2D electron density and mobility are identical in both crystallographic directions, even though we observed anisotropy in the undulation period and the strain relaxation in our samples. The average mobilities of sample E, obtained with the van der Pauw method and from Hall-bar devices, are in agreement to be within 10%.

The main reasons for the anisotropy of mobility reported in the literature are interface roughness, indium concentration modulation, asymmetric strain relaxation, anisotropic ordering effects, and anisotropic growth of dislocations [3,5,20,21,23,24]. A large anisotropy of mobility was observed in In_1−x_Ga_x_As QWs [3,20,21,23,24]. As demonstrated in Ref. [23], the anisotropy of mobility is strongly affected by indium compositional fluctuations in the In_1−x_Ga_x_As QWs system. Also, for 4 nm-thick InAs QWs, an anisotropy was reported [5]. Here, we employ 7 nm-thick QWs, which allows us to increase the fraction of the wave function contained in the binary InAs layer to approximately 70% [16] so that the electrons are less sensitive to the indium compositional fluctuations.

## 4. Conclusions

In this work, we have grown undoped InAs/In_0.81_Ga_0.19_As QW structures on GaAs(100) substrates using step-graded InAlAs metamorphic buffer layers. We have grown a series of samples with different overshoot layer thicknesses to tune the residual strain in the active region. We studied the influence of overshoot layer thickness on morphology, structure, strain, and electrical properties of the undoped InAs/In_0.81_Ga_0.19_As QW system. All samples show a crosshatch pattern oriented along the two orthogonal directions, $\left[1\bar{1}0\right]$ and [110], with an RMS roughness of the sample surface of (3.2 ± 1) nm. The residual strain was studied using XRD rocking curves and RSMs. Interestingly, we observed that the residual strain in the active region is compressive in samples with a 200 nm-thick overshoot layer and tensile in samples with thicker overshoot layers, confirming that the final strain state of the active region can be accurately tailored by the choice of overshoot layer composition and thickness.

In addition, the residual strain in the active region saturates to a constant value for overshoot layer thicknesses above 350 nm. We investigated how the state of relaxation of the active region influences the transport properties of the InAs QWs. The sample with a 400 nm-thick overshoot layer shows a peak electron mobility of 6.07 × 10^5^ cm^2^/Vs at 2.6 K with a carrier concentration of 2.31 × 10^11^ cm^−2^. Furthermore, the sample with the 400 nm-thick overshoot layer shows identical values of 2D electron density and mobility along both crystallographic directions. We attribute this to the strong confinement of the carriers to the inside of the binary InAs QW and a negligible effect of misfit dislocations, which are completely confined to the graded buffer layer. We believe that our study will promote further development of InAs/In_x_Ga_1−x_As QW systems for high-speed electronic devices.

## Figures and Tables

**Figure 1 nanomaterials-14-00592-f001:**
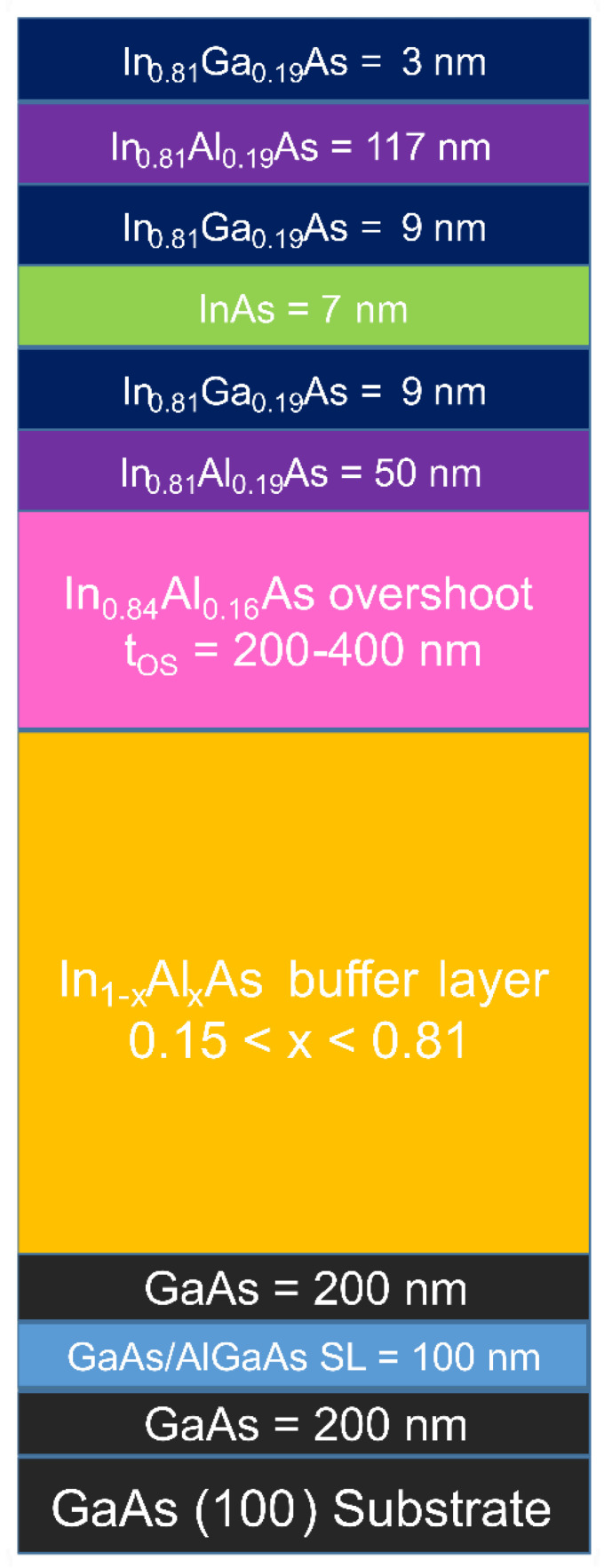
A schematic of the layer structure (not to scale).

**Figure 2 nanomaterials-14-00592-f002:**
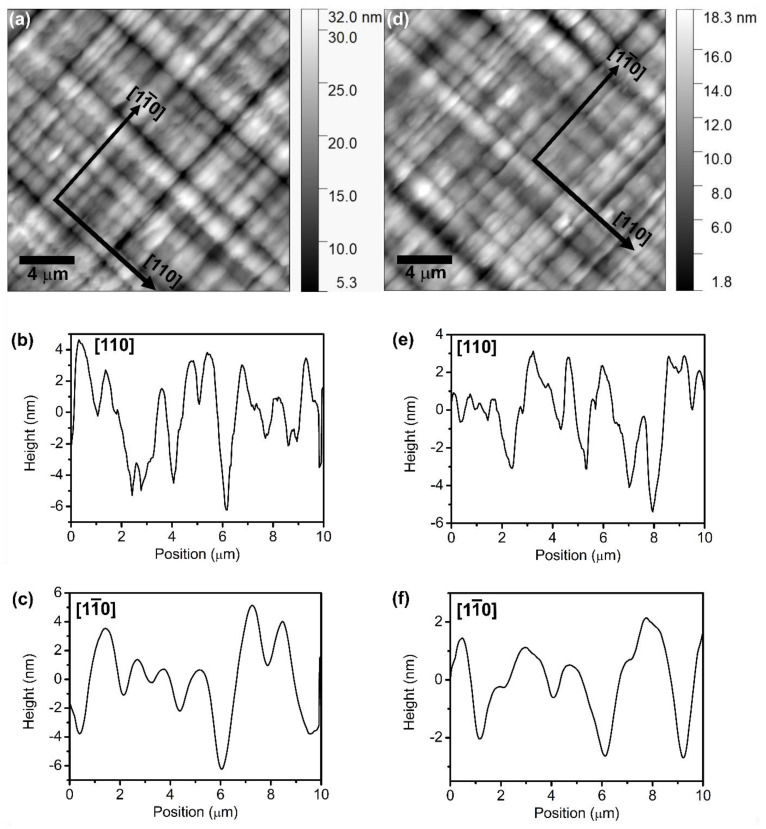
(**a**) 20 × 20 μm^2^ AFM image showing the surface morphology of sample A. (**b**) Height profile along the [110] direction of sample A. (**c**) Height profile along the $\left[1\bar{1}0\right]$ direction of sample A. (**d**) 20 × 20 μm^2^ AFM image showing the surface morphology of sample E. (**e**) Height profile along the [110] direction of sample E. (**f**) Height profile along the $\left[1\bar{1}0\right]$ direction of sample E.

**Figure 3 nanomaterials-14-00592-f003:**
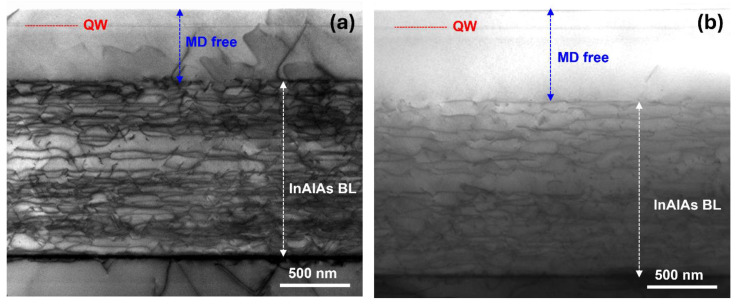
Cross-sectional STEM-HAADF images of (**a**) sample C; and (**b**) sample E.

**Figure 4 nanomaterials-14-00592-f004:**
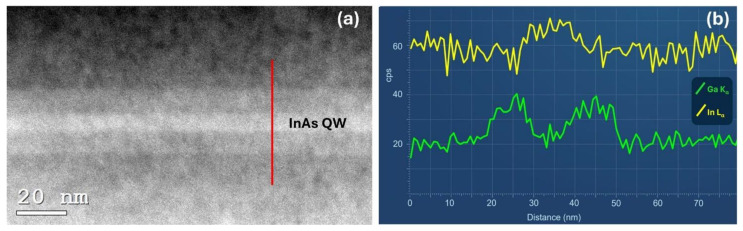
(**a**) STEM-HAADF image of the In_0.81_Ga_0.19_As/InAs/In_0.81_Ga_0.19_As layers of sample E. (**b**) EDS line scan obtained along the red line in panel (**a**).

**Figure 5 nanomaterials-14-00592-f005:**
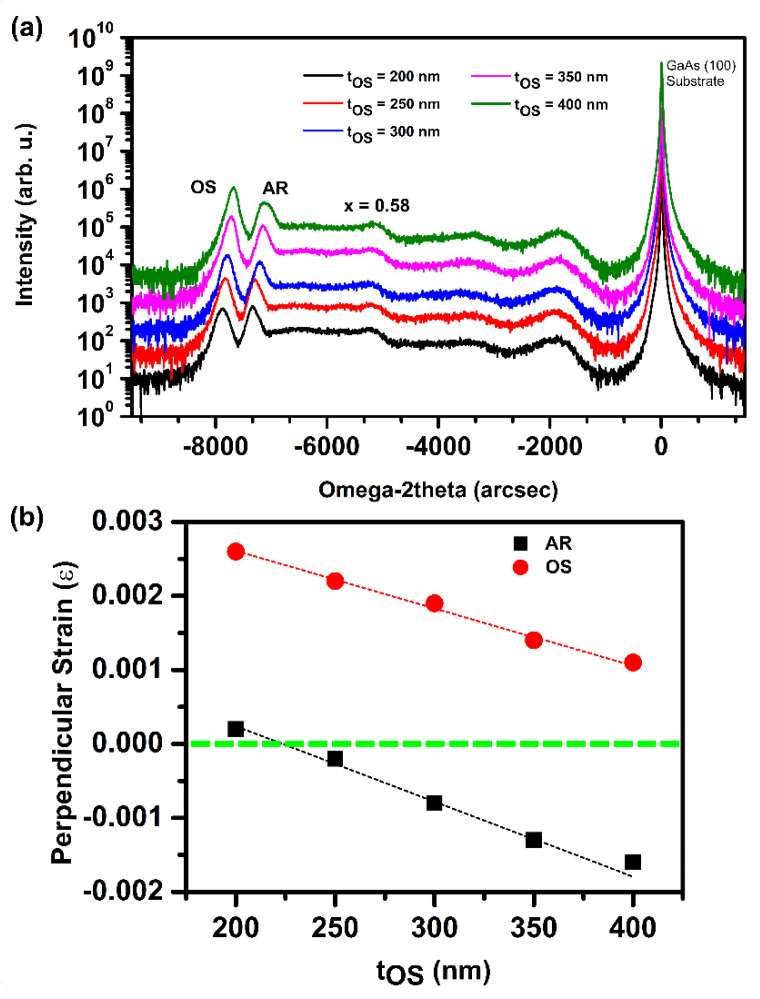
(**a**) (004) ω-2θ scans obtained along the 0° azimuth from a series of InAs/In_0.81_Ga_0.19_As QW samples with different overshoot thicknesses t_OS_. (**b**) Residual perpendicular strain in the active region (x = 0.81, black) and in the overshoot layer (x = 0.84, red).

**Figure 6 nanomaterials-14-00592-f006:**
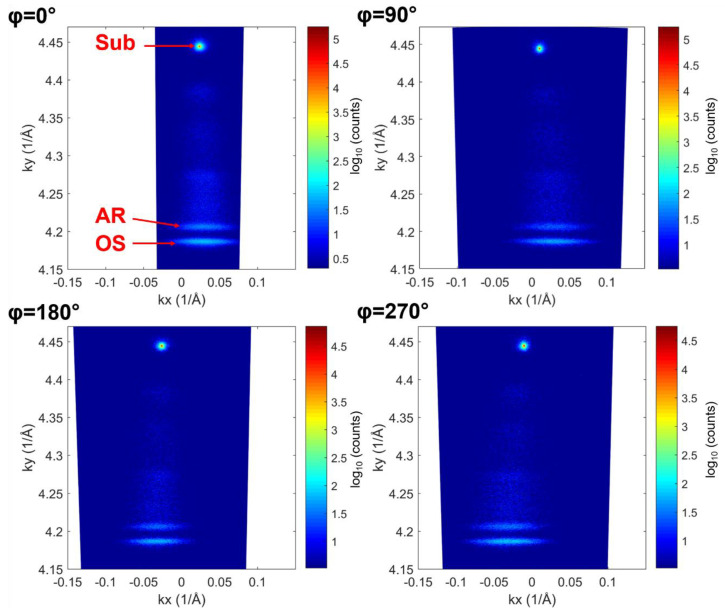
Symmetric (004) RSMs of sample E at φ = 0°/180° and 90°/270° sample azimuths, with scattering planes parallel to $\left[1\bar{1}0\right]$ and [110] directions, respectively.

**Figure 7 nanomaterials-14-00592-f007:**
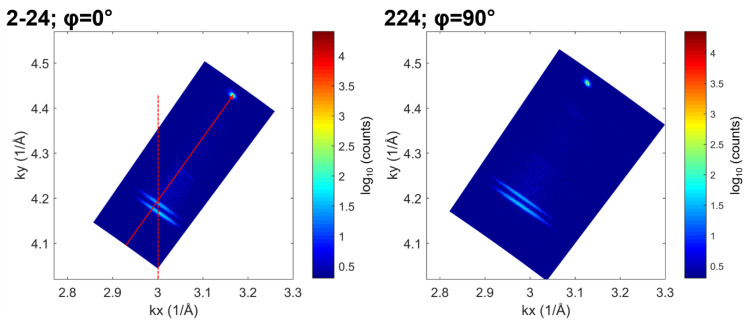
Asymmetric (224) RSMs of sample E at φ = 0° and 90° sample azimuths, with scattering planes parallel to $\left[1\bar{1}0\right]$ and [110] directions, respectively. In the φ = 0° RSM, the AR and substrate peaks are aligned along the (224) scattering vector (red arrow), and the AR and OS peaks are aligned along the vertical direction (dashed line).

**Figure 8 nanomaterials-14-00592-f008:**
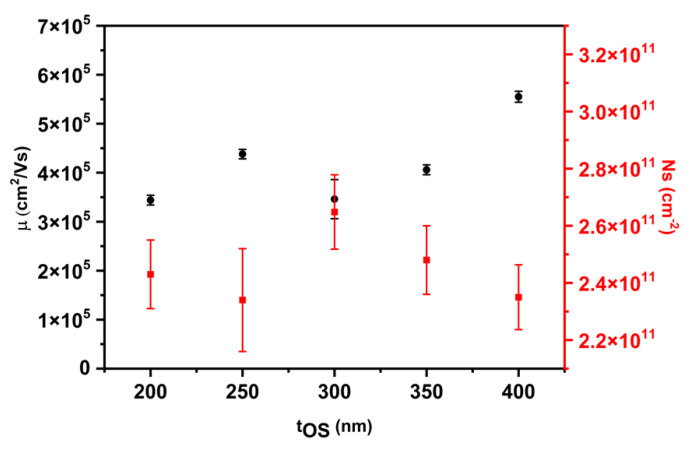
The 2D electron mobility µ (black symbols) and density N_s_ (red symbols) of the InAs/InGaAs 2DEGs at 4.2 K as a function of overshoot layer thickness.

**Table 1 nanomaterials-14-00592-t001:** Lattice parameters a, residual strain ε, and In fraction x_In_ in Sample E.

	Direction	*a^par^* (Å)	*ε^par^*	*a^perp^* (Å)	*ε^perp^*	*a^relaxed^* (Å)	*x_In_*
AR	$\left[1\bar{1}0\right]$	5.976	−0.0002	5.967	−0.0016	5.977	0.80
	[110]	5.996	0.0031				
OS	$\left[1\bar{1}0\right]$	5.976	−0.0025	5.995	0.0006	5.991	0.83
	[110]	6.000	0.0014				

## Data Availability

Data are contained within the article.

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
