# Peer review of "Influence of an Overshoot Layer on the Morphological, Structural, Strain, and Transport Properties of InAs Quantum Wells"

_nanomaterials, 2024, doi:10.3390/nano14070592_

Round 1
Reviewer 1 Report
Comments and Suggestions for Authors
In this work, the authors grew undoped InAs/In0.81Ga0.19As quantum well structures on GaAs(100) substrates by using step-graded metamorphic InAlAs buffer layers. To control the residual deformation in the active region, they used an overshoot layer. Quite detailed structural studies have been carried out. Electrical properties studies were carried out. The optimal overshoot layer value from the authors’ point of view was obtained.
While reading the work, the following questions arose:
1. The source of electrons in the system and how it depends on changes in the structure of the samples are not clear.
2. The explanation for the 1.5-fold increase in mobility associated with random deformation fields is, in my opinion, insufficient.
Please add a discussion of the effect of single-axis strain on the band parameters of InAs. In particular, a decrease in the effective mass will lead to an increase in mobility and a slight decrease in the concentration of charge carriers at a fixed value of the chemical potential.
Author Response
Review Report (Reviewer 1)
In this work, the authors grew undoped InAs/In0.81Ga0.19As quantum well structures on GaAs(100) substrates by using step-graded metamorphic InAlAs buffer layers. To control the residual deformation in the active region, they used an overshoot layer. Quite detailed structural studies have been carried out. Electrical properties studies were carried out. The optimal overshoot layer value from the authors’ point of view was obtained.
Reply: We thank the reviewer for the careful evaluation of our manuscript and the useful comments.
While reading the work, the following questions arose:
- The source of electrons in the system and how it depends on changes in the structure of the samples are not clear.
Reply: We agree that the source of electrons in the system was not specified. We have now added a sentence on page 3 to make this clear:
Although nominally undoped, electronic charge in the quantum well is due to a deep-level donor state in the InAlAs barrier band gap, whose energy lies within the InGaAs/InAlAs conduction band discontinuity [24a].
[24a] Capotondi, F.; Biasiol, G.; Vobornik, I.; Sorba, L.; Giazotto, F.; Cavallini, A.; Fraboni, B. Two-dimensional electron gas formation in undoped In0.75Ga0.25As/In0.75Al0.25As quantum wells. J. Vac. Sci. Technol. B 2004, 22(2), 702-706. https://doi.org/10.1116/1.1688345.
Since we have added a new reference, we have also updated the numbering of the references.
- The explanation for the 1.5-fold increase in mobility associated with random deformation fields is, in my opinion, insufficient.
Please add a discussion of the effect of single-axis strain on the band parameters of InAs. In particular, a decrease in the effective mass will lead to an increase in mobility and a slight decrease in the concentration of charge carriers at a fixed value of the chemical potential.
Reply: We agree with the possible explanation proposed by the reviewer for the observed behavior of the mobility and charge carriers. Therefore, we added this explanation to the text of the manuscript on page 11:
This indicates that a certain amount of tensile strain in the QW promotes mobility. Indeed, it is well known that the bandgap of InAs reduces under the influence of tensile strain [34,35]. Strain also affects the effective mass of the material [34,35]: due to the reduction of bandgap, the effective mass also decreases. Since carrier mobility is inversely proportional to the carrier effective masses, a reduction in the effective mass will lead to an increase in mobility.
[34] Pryor, C. E.; Pistol, M. E. Band-edge diagrams for strained III–V semiconductor quantum wells, wires, and dots. Phys. Rev. B 2005, 72(20), 205311. https://doi.org/10.1103/PhysRevB.72.205311.
[35] Jain, N.; Mal, I.; Samajdar, D. P.; Bagga, N. Theoretical exploration of the optoelectronic properties of InAsNBi/InAs heterostructures for infrared applications: A multi-band k· p approach. Mater. Sci. Semi. Proc. 2022, 148, 106822. https://doi.org/10.1016/j.mssp.2022.106822.

Reviewer 2 Report
Comments and Suggestions for Authors
Author Response
Review Report (Reviewer 2)
The authors investigate the influence of overshoot layer thickness on the morphology, electrical properties, and other characteristics of undoped InAs quantum wells on GaAs(100) substrates. The transition and buffer layers together provide a dislocation-free active region for InAs quantum wells. Variations in the thickness of the transition layer can cause changes in the residual strain properties of the active region.
Reply: We thank the reviewer for the careful evaluation of our manuscript and the useful comments.
(1) The composition and thickness of the overshoot layer indeed impact the strain state of the active region. Is In0.84Al0.16As the ideal composition for achieving the overshoot effect in the studied system? Any optimizing strategies?
Reply: It has been shown that for InxGa1-xAs quantum wells with x = 0.72, the optimal overshoot layer In concentration is between x = 0.76 and x = 0.80, see Capotondi et al., Thin Solid Films 484 (2005) 400. Here we study pure binary InAs quantum wells. This situation was discussed by Benali et al., J. Crystal Growth 593 (2022) 126768. They proposed to increase the thickness of the InAs quantum well to increase the fraction of the 2DEG density contained in the binary QW region from 45% to 69%. To sustain the additional strain created by making the InAs QW thicker, they increased the In composition of the InGaAs and InAlAs regions to 0.81. We follow the same strategy here, and the obtained results show that we are close to the ideal composition. In the future, the samples could be further optimized by following the approach of Capotondi et al., i.e., samples with different overshoot layer composition could be grown and compared.
(2) There are some typing errors that need to be carefully checked, such as ‘1-10’ direction.
Reply: We have fixed these typos. Instead of [1-10] direction we now write .
(3) What is the innovation of the article, which is not outstanding enough.
Reply: With respect to previous work, the two main novelties of our article are the use of reciprocal space maps for a three-dimensional strain mapping, and the investigation of a possible anisotropy in mobility. At difference to previous work, we do not observe an anisotropy, which we attribute to a major confinement of the carriers to the binary InAs QW. We have added a sentence on page 12 to make this novelty clearer:
Also, for 4 nm thick InAs QWs, an anisotropy was reported [5]. Here, we employ 7 nm thick QWs, which allows to increase the fraction of the wave function contained in the binary InAs layer to approximately 70% [16], so that the electrons are less sensitive to the indium compositional fluctuations.

Reviewer 3 Report
Comments and Suggestions for Authors
Dear authors,
Thank you for an interesting work with sound results clearly presented. I’m very happy to recommend your work for publication in Nanomaterials, after very few corrections aimed to further improve the presentation of your work:
1. Could you please clarify in text which surface of the sample was studied with AFM? Presumably it should be the capping layer, but I found no clear statement in the text concerning the question.
2. In the discussion of figure 5b (the last paragraph on the page 7), you state that the strain in overshoot layers becomes constant above 350 nm, and in a similar manner you state that in the active region the strain also saturates for overshot layer thicker than 350 nm. This is not obvious from the dependence shown on the figure 5b, and both dependences for the overshot layer and for the active region are quire linear for the whole region of thicknesses shown on the image. I would suggest depicting the linear trend on the graph for both dependences, then the small deviation towards saturation would become visible. In case you know about the saturation from other measurements you could also refer to them.
3. On the figure 6 the colour scale is missing a title and units, and I didn’t find them in text neither.
4. The title and units of colour scale are also missing on the figure 7 too.
Comments on the Quality of English LanguageNone, the English is good.
Author Response
Review Report (Reviewer 3)
Dear authors,
Thank you for an interesting work with sound results clearly presented. I’m very happy to recommend your work for publication in Nanomaterials, after very few corrections aimed to further improve the presentation of your work:
Reply: We thank the reviewer for the careful evaluation of our manuscript and the useful comments.
- Could you please clarify in text which surface of the sample was studied with AFM? Presumably it should be the capping layer, but I found no clear statement in the text concerning the question.
Reply: We thank the reviewer for highlighting this point. We have now added a sentence on page 4 to make this clear:
The surface morphology of all samples with different overshoot layer thickness was investigated by AFM. The AFM micrographs were acquired on the free surface of the samples (the capping layer of the samples) which reflects the morphology of the buried InxAl1-xAs/InxGa1-xAs interfaces.
- In the discussion of figure 5b (the last paragraph on the page 7), you state that the strain in overshoot layers becomes constant above 350 nm, and in a similar manner you state that in the active region the strain also saturates for overshot layer thicker than 350 nm. This is not obvious from the dependence shown on the figure 5b, and both dependences for the overshot layer and for the active region are quire linear for the whole region of thicknesses shown on the image. I would suggest depicting the linear trend on the graph for both dependences, then the small deviation towards saturation would become visible. In case you know about the saturation from other measurements you could also refer to them.
Reply: We thank the reviewer for carefully checking our manuscript and for noticing this point. We agree with the suggestion given by the reviewer. We have added the linear trend on the graph for the strain as a function of the overshoot layer in Figure 5(b). We updated figure 5 on page number 7 of the manuscript. We have also edited the text, to soften our statement:
The overshoot layers are compressively strained for all tOS, but with increasing tOS the strain decreases. and becomes constant above 350 nm. Instead, the strain in the active region switches from compressive to tensile and tends to saturate above 350 nm, where the overshoot layer is nearly relaxed.
- On the figure 6 the colour scale is missing a title and units, and I didn’t find them in text neither.
Reply: Thanks, we have updated Figure 6.
- The title and units of colour scale are also missing on the figure 7 too.
Reply: We thank the reviewer for this suggestion. We have updated the figure.

Round 2
Reviewer 3 Report
Comments and Suggestions for Authors
Thank you for corrections. I suggest to accept the paper in the present form